# Pediatric Urology Metaverse

Marcello Della Corte [1,2] , Erica Clemente [3,*] , Enrico Checcucci [4] , Daniele Amparore [1] , Elisa Cerchia [2], Berenice Tulelli [5], Cristian Fiori [1], Francesco Porpiglia [1] and Simona Gerocarni Nappo [2]

1   Division of Urology, Department of Oncology, School of Medicine, San Luigi Gonzaga Hospital, University of Turin, Regione Gonzole 10, 10043 Orbassano, Italy; dellacortemarcello@gmail.com (M.D.C.)
2   Division of Pediatric Urology, Regina Margherita Hospital, 10126 Turin, Italy
3   Department of Medical Sciences, University of Turin, Via Verdi 8, 10124 Turin, Italy
4   Department of Surgery, Candiolo Cancer Institute, FPO-IRCCS, 10060 Candiolo, Italy
5   Department of Pediatric Urology, Children and Mother Hospital, 69500 Lyon, France
*   Correspondence: clemente.erica@outlook.com; Tel.: +039-011-902-6477

**Abstract:** In the last decades, a digital revolution has transformed several aspects of people's lives worldwide. Consequently, many substantial changes have concerned numerous professional environments, including medical ones. Among all the different new instruments available in this field, the metaverse is the most futuristic one and seems to be likewise promising. The metaverse is an emerging resource in healthcare, resulting from the integration of virtual and physical reality. It is particularly valuable in surgical operations, since it allows surgeons to perfectly visualize patients' anatomy. Metaverse applications even include the pediatric field—in particular, the implementation of children and parents' shared decision-making processes, as well as prenatal diagnosis and fetal surgery. This resource further represents a rising opportunity in pediatric urology: the development of 3D virtual models and robotic surgery will allow surgeons to explore surgical fields, perfectionating their own professional skills. The metaverse will empower pediatric urologists, patients and their families in many ways, and each one of them deserves to be explored to the fullest. In this work, we aim to discuss the current applications of the metaverse in pediatric urology and its future perspectives.

**Keywords:** metaverse; UroVerse; augmented reality; robotic surgery; pediatric urology

## 1. Introduction

The metaverse is the fusion between real and virtual worlds, aimed to generate a three-dimensional (3D) digital reality to carry out any sort of human activity through the personification of characters via avatars. Although its main applications involve social, economic and cultural activities, recently, even healthcare services are finding the metaverse to be a new, stimulating opportunity [1].

The last decades have improved the spread of new technologies, involving healthcare systems among all the concerned areas. Healthcare services have recently encountered some significant system transformations, resulting in implemented informatic devices and digital sources.

COVID-19-related restrictions and the need to replan health welfare have led to more telemedicine taking place, forcing the development of new systems to promote remote health assistance. In particular, the remote collection of clinical data, medical images and vital signs has been fine-tuned, developing new informatic platforms and new medical tools [2].

A further leading actor, even if not properly born in the healthcare field, is artificial intelligence (AI). This technology, or 'property', can be defined as the development of algorithms that make machines able to transform themselves from executors to planners. AI commutates the computer into an electronic brain, enabling it to solve problems and

analyze words and concepts, thus performing cognitive functions as well as a human. In order to improve the progressive skills acquisition of electronic devices, machine learning (ML) uses computational algorithms for data processing while letting the machine develop proper 'reasoning' [3].

In the process of building a metaverse, four pivotal technologies have progressively affirmed their own role: virtual reality (VR), augmented reality (AR), mixed reality (MR) and extended reality (XR). VR creates a realistic setting where the user experiences a digital world. AR combines a real space with a virtual entity represented in two dimensions (2D) or three dimensions (3D). MR mixes two worlds, the real and the virtual one. XR is a non-univocally defined technology; it exists to integrate VR, AR and MR in order to implement a metaverse. In other terms, XR combines the healthcare industry with other major ones to develop a new industrial niche [4].

In this work, we suggest futuristic opportunities for the metaverse in a pediatric urology context, highlighting the current application of the metaverse both in pediatric settings and in urology.

## 2. Metaverse Driving Forces

The recent healthcare revolution has led to the role switching of information technologies. The previous 'just supportive' function of internet has turned into *infrastructure*, in the absence of which, a complete paralysis of healthcare services should be expected. The implementation of data quality—in particular, video and audio contents—changed into the necessity for prompt and faster sharing, stimulating the introduction of 5G and 6G wireless networks [4]. These new tools have allowed higher volumes of processed data per unit of time than the previous network platforms. The consequent anthropological impact has been significant; the same content is now enjoyable by different people, connected from various places, with latency-free and real-time interactions, as well as face-to-face meetings.

Additionally, 3D systems have played a crucial role in this field. Their application in meeting platforms has reconstructed the medical setting, giving patients the subjective perception of being in an outpatient clinic. The physician side is professionally more exciting, due to the 3D reconstruction of organs and surgical spaces that allows a suggestive virtual immersion in human anatomical environments [2].

Lastly, the difficulties in accessing healthcare related to COVID-19 have acted as a strong driving force in informatic skills acquisition, both by healthcare professionals and by the general population [4]; paradoxically, the gradual overcoming of distances has progressively reduced physical separation, and wireless connections act as conduction cables.

## 3. Metaverse Applications in Healthcare Industry

A complete meta-analysis of the metaverse in the healthcare industry has been proposed by Chang Wong Lee [4]. His work focused on three macro areas where the metaverse is gaining ground: surgical operations, healthcare education and treatment.

Surgical operations represent the leading field where the metaverse can give its best; the implemented visualization of anatomy minimizes the complications and maximizes the surgical accuracy [5].

Offering a multi-sense experience, the metaverse can revolutionize healthcare education, both for beginners and for the most skillful physicians. The 3D visual experience stimulates a better comprehension of anatomy, and complex surgical operations become more understandable.

Lastly, the metaverse's potential key role in treatment strategies will lead to its assertion in pain management [6], especially in chronic diseases or in cognitive rehabilitation [4,7].

Many medical specialties have been exploring metaverse potentials. In his meta-analysis, Lee [4] identified 15 application areas; nevertheless, this number is certainly going to grow.

## 4. Pioneering Applications of Metaverse in Pediatrics

Several attempts of metaverse utilization have regarded pain relief in outpatient clinics. The use of virtual reality offers an alternative strategy to ensure a funny distraction, resulting in painless and anxiety-free experiences during medical visits and invasive procedures [8].

A recent study by Moon et al. [9] reported the therapeutic effects of metaverse-based rehabilitation. The authors enrolled 26 children with cerebral palsy into a randomized controlled trial, demonstrating that metaverse-based physical therapy is more effective than conventional ones.

Lastly, Chang et al. [10] provided the first interactive multimedia mixed reality game with the purpose of implementing shared decision-making processes in children with atopic dermatitis. In chronic diseases, such as atopic dermatitis itself, the issues in children's communication and their difficulties in explaining their own clinical symptoms limit the treatment's effectiveness; therefore, therapeutic decisions are often decided with the children's parents' collaboration. The use of the metaverse in shared decision making has been validated by the System Usability Scale and is useful in clinical practice.

## 5. Educational Applications

The COVID-19 pandemic complicated medical training worldwide. To overtake this limit and ensure high standards of surgical training, a virtual platform was developed, with more than 200 thoracic surgeons join the operating room in the metaverse. This program, launched by SNU Bundang Hospital, consisted of a smart operating room virtually reproduced with 3D glasses available for surgeons [11]. Even if still unknown in pediatric urology, an approach such as that would improve the acquirement of surgery skills even in the robotics field, which shows a slow learning curve [12]. Additionally, the metaverse would help both students and beginners gain a better understanding of complex fetal malformations.

Furthermore, a useful application of the metaverse may concern health education purposes for patients and for their caregivers. Digital devices and media are the preferred channels that people generally use for educational or information scopes [13], and the metaverse creates an immersive environment where real and virtual are mixed and their combination is simplified [2]. Since a preliminary and promising experience has already been implemented in oral hygiene education [8] several applications may be considered even in different fields, such as pediatric urology.

First of all, the metaverse can be applied to the virtual teaching of how to manage a stoma or other urinary diversions, for example, the correct execution of clean intermittent catheterization [14].

In complex cases, multidisciplinary meeting in the metaverse may be hypothesized, ensuring high standards of quality thanks to the full involvement of patients and families and also allowing the participation of experienced specialists connected from distant places.

## 6. Perioperative Support

Despite the proper accuracy in performing presurgical counseling, the patients and—especially in pediatric settings—their parents often fail to retain all the offered information. The difficulties in approaching medical and surgical notions for an uneducated individual may result in dissatisfaction and frustration. The progression of conventional practices is ceding to virtual reality technologies. Starting from oncological pediatric surgery, three-dimensional virtual models (3DVMs) can improve the parental understanding of surgical management [15]. In this sense, even AI can exert a prominent role in radiological data processing—in particular, when many dubious radiological patterns are not easily recognizable with the human eye. AI can be potentially set to analyze preoperative diagnostic images to predict potential intraoperative difficulties and to suggest the best surgical strategies [3]. Images may be shown with a two-dimensional (2D) flat screen or in a 3D virtual

environment using a head-mounted device. When the metaverse is explored by several people, each one meets the other ones' avatars, also called 'digital twins' [16] (Figure 1a,b).

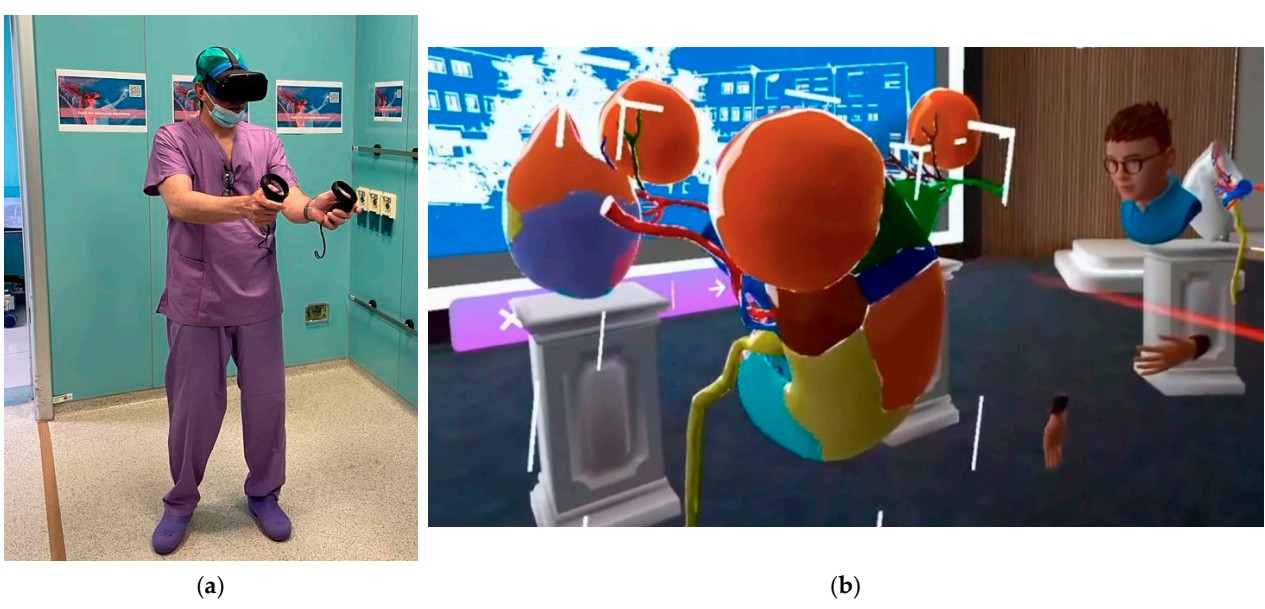

(**a**) (**b**)

**Figure 1.** (**a**) The surgeon explores the field pre-operation thanks to a head-mounted device and two handgrip tools. (**b**) His digital twin observes a 3D virtual model.

Images magnification, through virtual reality technology, creates a multidimensional setting that completely absorbs the spectator's different senses. The resulting environment is a fusion between the virtual world and the real one, since the surgeon can interact with it through body movements [16]. The metaverse projects the spectator into the virtual operating theater, offering a new important tool for presurgical planning and counseling, with the additive values of overcoming distance and possible home usage [17].

## 7. Fetal Urology

Ultrasonography (US) and magnetic resonance imaging (MRI) represent the key technologies in assessing fetal anatomy. Currently, 3D-US is flanking the less recent ones as an additional method to evaluate fetuses when anomalies are suspected. This technology exploits a sophisticated digital reconstruction technique and also helps to illustrate malformations during parent counselling [18].

The use of the metaverse in fetal medicine, introduced by Werner et al. [19], depicts a promising area of application. The COVID-19 pandemic has forced the development of technological tools to train specialists in multidisciplinary discussions, overcoming geographical and logistical issues.

The same technology has also allowed the possibility of displaying shared 3D navigation inside an organ. The 3D reconstruction of MRI images allows navigating inside of a virtual model thanks to VR. Surprisingly, the organs themselves can become the environment of the 3D virtual meeting.

In this setting, AI may help in recognizing typical or atypical malformative patterns, especially when the smallest pathological signs are not recognized by the human eye [3]. Therefore, a higher diagnostic sensibility will be developed, and the metaverse will be the best setting to explore and to learn the newly identified anatomical findings.

The near future is expected to show the application of the metaverse in prenatal parent counseling. Thanks to some head-mounted devices, parents will be immersed in a virtual world, ensuring a full comprehension of the anatomical defects, empowering them in a multidisciplinary discussion and improving their decision making.

## 8. UroVerse and PedUroVerse

In the last few years, the fusion between 3DVMs and robotic platform images has led to the development of 3D augmented reality-guided surgery. The application of this technology, chiefly in prostate and kidney cancer surgery, improves the identification of landmarks, vessels and nerve pedicles, helping in many critical moments of both the demolitive and reconstructive phases of surgical procedures, even resulting in kidney transplantation to identify atheromatic plaques [16,20,21].

In particular, 3DVMs allow GPS-like surgical navigation strategies and elevate the quality of traditional intraoperative ultrasounds or fluorescence imaging [22].

The combination of 3DVMs, VR, AR, MR, XR and their related mobile applications has been shifting the surgical paradigm. The application of the metaverse in neurology and neurosurgery made Kundu et al. coin the word "NeuroVerse", which implementation can raise invasive procedure quality, both surgical and interventional, pursuing higher standards during outpatient visits and medical care and redefining surgical practices and training [7].

Since the metaverse is a 3D virtual world resulting from the fusion between the real and the virtual ones, the possibility to create innumerable virtual worlds implies the potential development of as many metaverses. On the heel of Kundu et al. [7], we should admit that the "UroVerse" has been defined [2]. In particular, the assistance of 3DVMs has recently offered telementoring support during a partial nephrectomy among three different countries. A 3DVM was displayed inside the robotic console while another surgeon manipulated it during the intervention from another place. The 3DVM is useful in perioperative surgical planning; the surgeons interact with the model in the metaverse by using head-mounted displays [16,23].

Thanks to the development of the newest touchless devices (Ultraleap Ltd., 2522 Leghorn Street, Mountain View, California, 94043, USA), surgeons can move in the metaverse without any additional wearable tools [24]. The presence of proper sensors enables the software to read the user's gestures, rendering their movements virtual through a codified system. In this way, 3DVMs can be useful even during surgery, not needing to respect sterility requisites (Figure 2).

We hypothesize that the near future will similarly propose "PedUroVerse" opportunities, which means several possible applications of the metaverse in pediatric urology.

Some anticipators have already been proposed in the last couple of years—in particular, since the UMBRELLA SIOP Guidelines admitted nephron-sparing surgery (NSS) in particular cases of Wilms tumors [25]. The NSS approach aims to preserve the maximum amount of renal parenchyma to prevent renal failure, especially in cases of bilateral tumors or future chemotherapy needed, due to its nephrotoxicity. Three-dimensional printing has contributed to preparing surgeons for NSS, ensuring an optimal pre-operative comprehension of the surgical dissection plans [26]. Similarly, 3D augmented reality will guide future research perspectives in pediatric urology, and partial nephrectomies have already been performed thanks to this technology [27].

Undoubtedly, robotic surgery remains the ideal candidate to accommodate the metaverse in daily clinical practice. In pediatric urology, robotics is progressively gaining its own role. While pyeloplasty for ureteropelvic junction obstruction remains the commonest procedure executed with this approach, other operations have progressively adapted, such as extravesical ureteral reimplantation, bladder neck reconstruction, stones removal and the Mitrofanoff procedure.

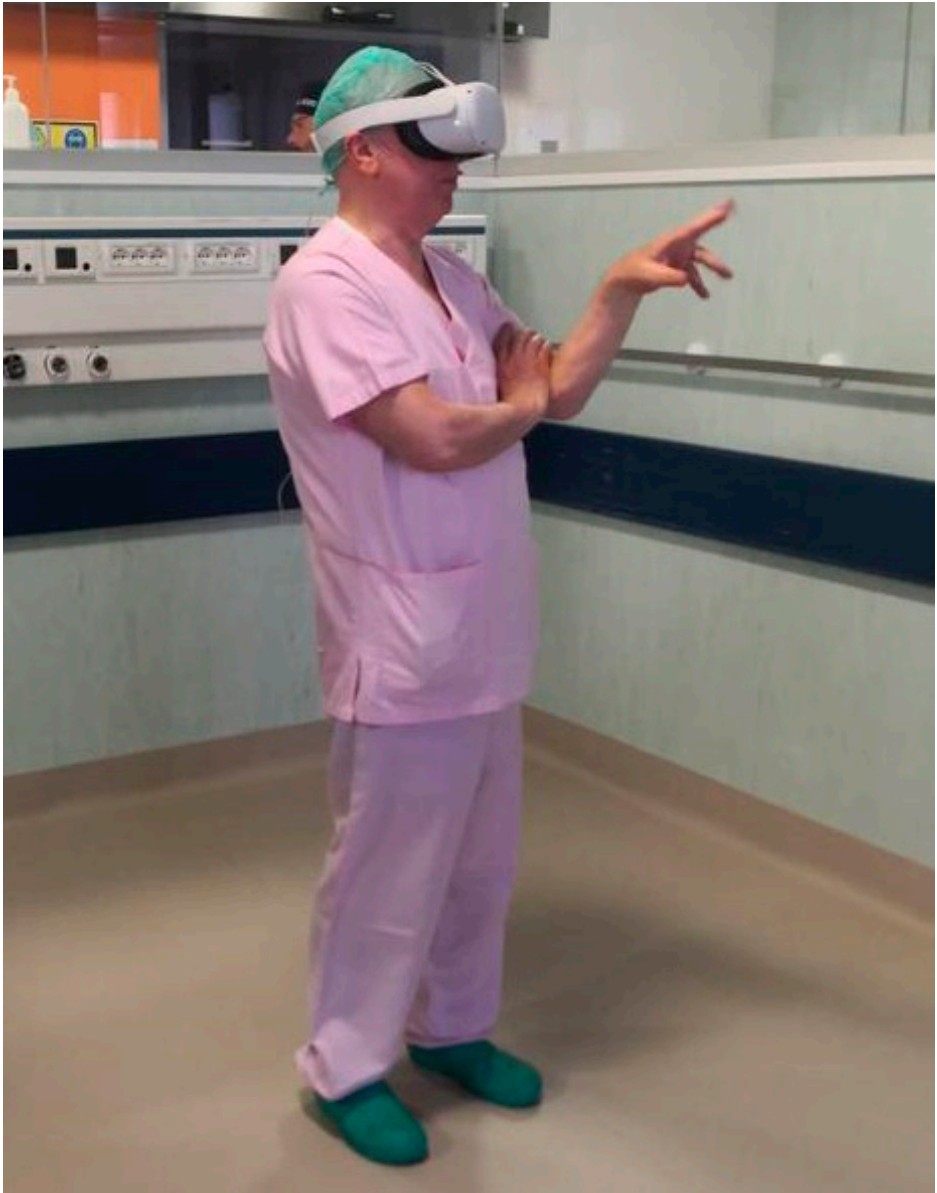

**Figure 2.** The surgeon interacts with the 3DVM of the kidney he is going to operate on. The sensors read his movements without requiring wearable tools.

Although laparoscopy remains the preferable choice for benign indications or uretero-pelvic junction obstruction (UPJO) in small infants, in renal surgery, robotics improves the safety of vessel isolation, the management of thrombus veins and nephron-sparing techniques in oncological settings. Big renal cysts are more easily manageable with robots, both if placed laterally on the upper pole and thanks to the assistance offered by ICG fluorescence that helps in identifying the cyst wall.

Robotics is still useful in bladder stone removal and facilitates the reconstruction of the urinary tract [12]. In all these situations, 3DVMs would facilitate an intraoperative approach. In renal surgery, 3DVMs may flank ICG fluorescence, maximizing the precision in the localization process of a cyst or a tumor mass, and they would overcome the intraoperative ultrasound guidance for robotic renal stones management (Scheme 1).

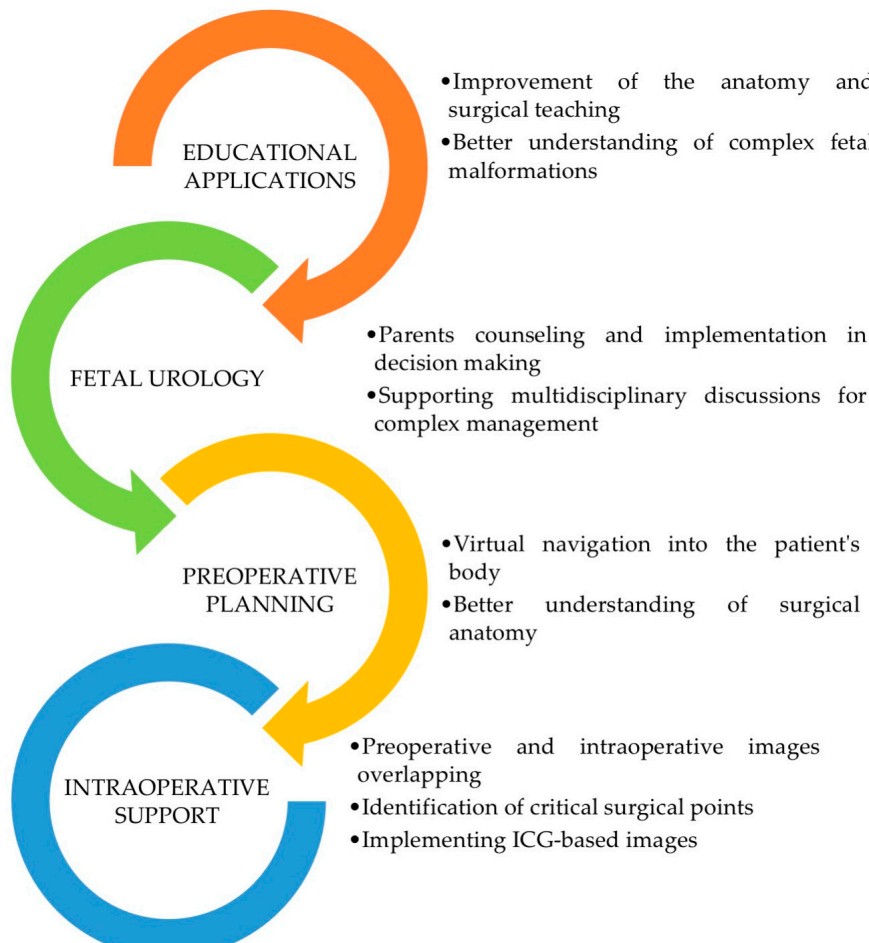

**Scheme 1.** PedUroVerse applications.

AI can offer in all these cases full support to improve the intraoperative information available to the surgeon. The minimally invasive approach deprives the surgeon of tactile feedback. Through the ML process, AI would progressively learn the effects that each robotic or laparoscopic maneuvers can exert on human tissues, then help the surgeon to avoid any surgically unpredictable damages, such as organ or vessel injuries [3].

Nevertheless, all that glitters is not gold. The maximization of these technological concepts in soft tissue surgery cannot ignore the limits due to tissue movements—in particular, the ones due to respiratory motions. Furthermore, a discrepancy between presurgical reconstruction based on MRI and CT scans and the actual intraoperative images could determinate some issues. In fact, medical imaging usually adopts a supine patient position, while, in urologic surgery, the flank one is preferred. In this way, a change in the spatial relationships easily takes place, as well as pneumoperitoneum-related additional deformations of the abdominal wall that may lead to a further loss of preoperative landmarks, including creating some artifacts until real-time correction [22]. For these reasons, new software for adaptable and prismatic algorithms are required.

Lastly, the high firmness of robot optics seems not to promise any involvement in the process of laparoscopy, which is conversely an optical system based on assistant surgeon manipulation, resulting in a critical sensitivity to minimal swinging movements that would reduce the operation accuracy.

### 9. Limits and the Bottleneck Effect

Although the metaverse is virtual, its potential limits are effectual. The metaverse is a complex reality; therefore, it cannot be labeled as the solution for all critical issues of surgery, robotics and healthcare systems.

The assumption of large-scale technology availability could represent a critical point [6]. In particular, it may represent an easy access to metaverse fruition for physicians working in centers with higher resource availability, instead of peripheral and smaller ones.

Similarly, the risk of being cut off from metaverse advances is concrete for poorer population groups, less-educated ones and elderly people. Lacking finance availability and specific competencies, these minorities would experience a 'digital divide' [6].

AI applications cannot replace surgeon skills. Human beings think and guess, while machines learn and combine. Human beings comprise and feel, while machines analyze and build. Human beings have potentially unlimited cognitive power, but machines are faster. These different properties can improve the global healthcare field worldwide when in synergy, but they cannot replace each other.

Furthermore, the metaverse's Achilles' heel is the high load of sensitive data that it carries around: privacy policies, regulations, cybersecurity, phishing and firmware are only few of the numerous legal and safety issues to be added to the metaverse vocabulary and are worth being deeply analyzed [16].

These aspects constitute the bottleneck effect that may inevitably affect the metaverse diffusion, reducing its power in temporal and spatial frames (Figure 3).

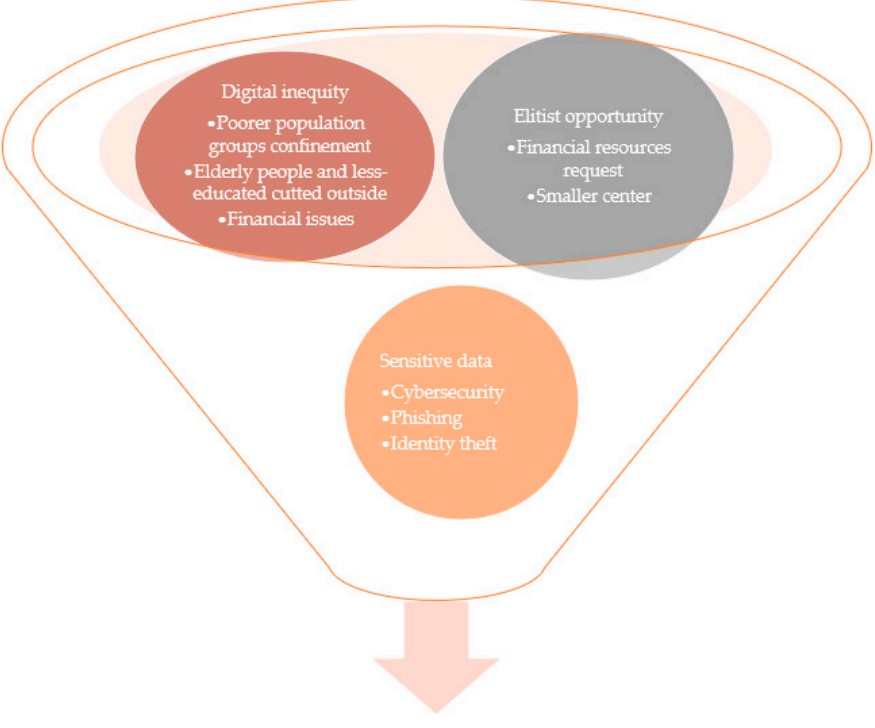

**Figure 3.** The 'bottleneck effect' is an expression that refers to issues that may impact metaverse diffusion. The figure summarizes their main characteristics.

Lastly, an already known consequence of prolonged screen exposure in childhood and adolescence is related augmented incidences in mental health issues and myopia [6].

## 10. Conclusions

The metaverse is an emerging opportunity for healthcare services. Its preliminary applications in urology seem to reserve it a potential role in the pediatric urology field, codifying the new PedUroVerse paradigm.

The PedUroVerse is improving surgical robotic possibilities, enlarging and perfectionating robotic applicability in pediatric scopes. In clinical outpatient settings, the metaverse will simplify doctor–patient communication, allowing counseling relationships both in perioperative and in prenatal diagnostic environments. Lastly, the metaverse is going to revolutionize even social and educational channels, implementing surgical teaching and promoting high-quality participation in multidisciplinary pediatric discussions. Many critical issues will need to be fixed, from legal policies to economic disparities.

We believe that the metaverse is a real and promising opportunity in pediatric urology to empower both physicians and patients, as well as their families. Currently, it remains an unexplored field that may use the many advantages seen in the experiences achieved by adult patients so far.

**Author Contributions:** Conceptualization, M.D.C., E.C. (Erica Clemente) and S.G.N.; methodology, M.D.C., E.C. (Erica Clemente), E.C. (Elisa Cerchia), B.T. and S.G.N.; software, M.D.C., E.C. (Erica Clemente) and S.G.N.; validation, S.G.N.; formal analysis, E.C. (Erica Clemente), E.C. (Elisa Cerchia), B.T. and S.G.N.; investigation, M.D.C., E.C. (Erica Clemente), E.C. (Elisa Cerchia) and S.G.N.; resources, M.D.C., E.C. (Erica Clemente), E.C. (Elisa Cerchia), B.T. and S.G.N.; data curation, M.D.C., E.C. (Erica Clemente) and S.G.N.; writing—original draft preparation, M.D.C., E.C. (Erica Clemente), E.C. (Enrico Checcucci), D.A., E.C. (Elisa Cerchia), B.T., C.F., F.P. and S.G.N.; writing—review and editing, M.D.C., E.C. (Erica Clemente), E.C. (Enrico Checcucci), D.A., E.C. (Elisa Cerchia), B.T., C.F., F.P. and S.G.N.; visualization, M.D.C., E.C. (Erica Clemente), E.C. (Enrico Checcucci), D.A., E.C. (Elisa Cerchia), B.T., C.F., F.P. and S.G.N.; supervision, E.C. (Enrico Checcucci), C.F., F.P. and S.G.N.; project administration, M.D.C. and S.G.N.; and funding acquisition, E.C. (Erica Clemente). All authors have read and agreed to the published version of the manuscript.

**Funding:** This research received no external funding.

**Institutional Review Board Statement:** Not applicable.

**Informed Consent Statement:** Not applicable.

**Data Availability Statement:** Not applicable. No new data were created.

**Conflicts of Interest:** The authors declare no conflict of interest.

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
