# Peer review of "Pediatric Urology Metaverse"

_2673-4095, doi:10.3390/surgeries4030033_

Round 1

Reviewer 1 Report

More on the dove-tailing influence of Artificial Intelligence to get to Augmented Reality would be helpful for most of us who are only getting up to speed with AI in the last few years

Author Response

Dear Reviewer,
We sincerely thank you for evaluating and reviewing our manuscript.
Thank you for your useful suggestion. The artificial intelligence is progressively changing the practice of surgery, improving the human skills in imaging, navigation and robotics. We believe that this concept is strongly pertinent with our manuscript. Therefore, we deepened the influence of artificial intelligence on augmented reality. In particular, we wrote new paragraphs into the sections “introduction”, “perioperative support”, “fetal urology”, “UroVerse and PedUroVerse” and “Limits and the bottleneck effect”.
We hope you will approve the additions. 
We remain at your disposal for any further improvement required.

Thank you again,

Best Regards

Reviewer 2 Report

Thank you for this opinion paper, Opinion, Pediatric Urology Metaverse. It is well written and illustrated. There are no further comments

Author Response

Dear Reviewer,
Thank you for reviewing our manuscript.
We sincerely thank you a lot for your considerations. 

Best Regards

Reviewer 3 Report

Authors should be congratulated for the topic addressed. The review is well-structured and easily readable. The role of augmented reality in pediatric urology was outstanded by Metaverse, which the authors highlighted properly. The significant content has several strengths: the comprehensive perspective, the clinical implication, and the citation of daily life experience that enhanced the originality of the manuscript.

Author Response

Dear Reviewer,
Thank you for reviewing our manuscript.
We sincerely thank you a lot for your considerations. We are so proud to read your review report. 

Best Regards

Reviewer 4 Report

In this narrative, the authors aim to analyze the futuristic opportunities that metaverse could get in the pediatric urology context, highlighting the current application of metaverse both in pediatric settings and in urology.

This study has some clinical significance but there are several considerations regarding the methodology. 

This is at best a narrative review where the authors have cherry-picked the articles of interest to suit their conclusions. It would have been ideal to proceed with a scoping or a systematic review - where all eligible studies should have been selected. 

The authors subjectively describe the results from the studies, instead, the authors should provide the actual numbers to highlight the difference in effect sizes, especially in the very few RCTS using metaverse in pediatric urology, when describing the outcomes of interest. It is unclear as to what the outcomes were in all studies assessed either - but a generic statement that metaverse has performed better were added.

This adds limited novelty to the existing literature. There is very high rate of self citation.

Sufficient to understand and interpret the written context. Minor grammatical errors. 

Author Response

Dear Reviewer,
We sincerely thank you for evaluating and reviewing our manuscript.
Thank you for your suggestions.
We are afraid that you probably did not consider the manuscript type which, in this case, is 'Opinion'. In particular, you mentioned the word 'methodology', but there are currently neither guidelines nor statements on how-to-write an opinion article, except for some suggestions on various scientific journals websites (e.g. MDPI Guidelines). The objective of an opinion article is to totally reflect the Authors’ point of view, even cherry-picking the articles of interest to suit Authors conclusions. This is the literal meaning of “opinion”. The scoping or systematic review would not have reflected both the objective and the purposes of the Authors.

Since our work aims to describe an undiscovered and unexplored field, the suggestion of Review is not perfectly coherent. 

Thank you again for your support.

Best Regards

Round 2

Reviewer 4 Report

Since it is just an opinion piece - I would be fine with the revised manuscript. It is still not adding much to the existing literature. 

Appropriate, requiring some minor editing.